# Hindbrain Stimulation Modulates Object Recognition Discrimination Efficiency and Hippocampal Synaptic Connections

**DOI:** 10.3390/brainsci13101425

**Published:** 2023-10-07

**Authors:** Alesha Heath, Michelle Madore, Karina Diaz, M. Windy McNerney

**Affiliations:** 1Department of Psychiatry and Behavioral Sciences, Stanford University School of Medicine, Stanford, CA 94305, USA; 2Department of Veterans Affairs, Sierra-Pacific Mental Illness Research Educational and Clinical Center, Palo Alto, CA 94304, USA; 3Department of Pharmacology and Physiology, Graduate School of Arts and Sciences, Georgetown University, District of Columbia, Washington, DC 20057, USA

**Keywords:** repetitive transcranial magnetic stimulation, neuromodulation, hindbrain, synaptic plasticity, processing speed

## Abstract

(1) Background: The cerebellum is well known to have functionalities beyond the control of motor function. However, brain stimulation studies have not explored the potential of this region to impact downstream processes which are imperative to multiple neurological conditions. Our study aimed to look at preliminary evidence that hindbrain-targeted repetitive transcranial magnetic stimulation (rTMS) in mice could alter motor, cognitive and anxiety measures; (2) Methods: Male B6129SF2/J mice (*n* = 16) were given rTMS (*n* = 9) over lambda at 10 Hz for 10 min or Sham (*n* = 7) for 14 consecutive days. Mice then underwent a battery of behavioral measures. (3) Results: In the object recognition test, only rTMS-treated mice distinguished between the novel object at 5 min, whereas those that received Sham treatment continued to improve discrimination from 5 to 10 min. Additionally, over the 10 min test phase, rTMS-stimulated mice explored the objects less than the Sham mice. This was accompanied by increased colocalization of presynaptic and postsynaptic markers in the hippocampus in the rTMS mice (4) Conclusions: Hindbrain rTMS stimulation elicits improved processing speed in the object recognition test via structural plasticity mechanisms in the hippocampus and could provide additional ways of targeting these important substructures of the brain.

## 1. Introduction

Although highly regarded as a controller of managing somatic motor function, the connectivity and functional impact of the cerebellum to cognitive and emotional processes has been demonstrated through multiple studies [1,2]. There is significant bidirectional connectivity between the cerebellum and regions of the brain that are involved in emotional processing and cognition such as the prefrontal cortex and hippocampus [3,4]. Downstream neurotransmitter signals from the cerebellum directly affect these regions and the processes involved [4]. Additionally, multiple neuropsychiatric conditions show cerebellum alterations [1], which has gained more interest recently for the potential to alter these disorders through this brain region.

Repetitive transcranial magnetic stimulation, rTMS, is a popular neuromodulation method that has been successfully used to alter neuronal excitability [5]. rTMS is known to induce both short-term changes to neurotransmitters and synaptic plasticity, and long-term alterations to connectivity [6]. For clinical use, high-frequency rTMS is FDA approved for the treatment of depression [7,8], obsessive compulsive disorder [9], smoking cessation [10] and anxiety with comorbid depression [11]. There is also increasing evidence that rTMS intervention can have promising effects in other conditions including rehabilitation post stroke, schizophrenia, alcohol use disorder, Parkinson’s disease and Alzheimer’s disease [12], all of which primarily target cortical areas, especially the dorsolateral prefrontal cortex (DLPFC). However, we have limited knowledge of the effectiveness of rTMS in targeting other regions of the brain. Investigations into the biochemical pathways and behavioral impacts of rTMS from regions other than the DLPFC may uncover a target that is more efficacious for different neurological conditions. As the cerebellum is accessible to rTMS and is highly connected throughout the brain, this serves as a novel target with rehabilitation potential.

There have been some past studies to examine the effect of rTMS targeted to the cerebellum. In humans, cerebellum stimulation alters cortical motor-evoked potentials [13], improved outcomes in Cervical Dystonia patients [14] and improves social cognition [15]. In rodents, four weeks of rTMS directed at the cerebellum increased the size of the Purkinje cell dendritic tree and improved spatial memory performance, and in another experiment, two weeks of cerebellar rTMS promoted vesicular glutamate transporter 2–positive neuronal reinnervation after olivocerebellar lesion [16,17]. Though these studies have predominantly focused on the cerebellum’s involvement in movement and motor function, they demonstrate the downstream cortical effects rTMS can have when targeting this area.

We plan to expand on these studies by relating behavior to the hippocampal syntapto-generating effects of rTMS delivered to the hindbrain and cerebellum. As the hippocampus is a region that readily undergoes synaptic plasticity in support of memory formation throughout the lifespan [18], it serves as a critical region to target for the support of memory function. However, the induced magnetic field from rTMS is too superficial to reach the hippocampus itself [19], so relying on interconnected structures remains a viable option. The cerebellum is integral for memory and hippocampal function [20] but is an underexplored target region for brain stimulation.

In this preliminary study, we used a mouse model of rTMS therapy to target the hindbrain and assess the effect of two weeks of rTMS stimulation on motor, cognitive and anxiety measures in the mice. We also assessed whether stimulating the cerebellum could cause downstream changes to synaptic plasticity in the hippocampus, and inhibitory and excitatory connections in the frontal cortex. Thus, the current study aims to establish a functional relationship between hindbrain stimulation and hippocampal synaptic formation in terms of behavioral outcomes.

## 2. Materials and Methods

A total of 16 male B6129SF2/J mice aged 2–4 months were used for this study. All mice lived under a 12 h light/12 h dark cycle with free access to dry feed and water. All animal care and procedures complied with the Animal Welfare Act and were in accordance with institutional guidelines and approved by the VA Palo Alto Committee on Animal Research.

A timeline of experimental events can be found in Figure 1. Each of the mice underwent surgery to attach a coil support that allowed for consistent rTMS stimulation of the awake animals. Details about the surgical procedure can be found in Madore et al. [19]. In brief, animals were anesthetized with 3% isoflurane and maintained at 2–3% with a nose cone. After shaving and cleaning the site for sterility, a sagittal incision of approximately 10 mm in length was made with a scalpel blade. The periosteum was gently scraped away, and a premade coil support was attached at lambda with cyanoacrylate and reinforced with dental cement. The wound was then sutured around the coil support.

After five days of recovery from the coil support surgery, the mice were habituated to a separate stimulation cage and coil procedures for three days. Mice were randomly divided into receiving active rTMS stimulation (TMS *n* = 9) or inactive SHAM stimulation (SHAM *n* = 7). The mice were stimulated with 10 Hz for 10 min, for a total of 6000 pulses once a day, for 14 days with the TMS coil clipped to the coil support and the mice awake and freely moving.

At the end of the 14 days, the mice underwent three days of behavior testing. On the first day, the mice underwent the Open Field Test (OFT) for 10 min, which also served as the habituation for the Object Recognition Test (ORT). The second day the mice underwent the ORT. Mice were given ten minutes to explore two of the same objects in the same OFT arena, then three hours later tested for their object recognition memory with one familiar object and one novel object. We assessed the exploration of the objects at both five and ten minutes. On the final day the mice, underwent the O maze test for five minutes to assess anxiety phenotypes.

The following day, mice were transcardially perfused with 50 mL of cold 1 M PBS. Brains were immediately postfixed in 4% paraformaldehyde solution at 4 °C and transferred to 30% sucrose in PBS at least 48 h before cryosectioning into 30 μm sections. Free-floating sections were blocked and permeabilized with 10% goat donkey serum in 0.1% Triton in 1 M PBS and then incubated for 48 h with the rabbit polyclonal Postsynaptic density protein 95 (PSD95, 1:250, Thermo Scientific, Waltham, MA, USA) and mouse monoclonal Synaptic vesicle glycoprotein 2A (SV2A, 1:250, Santa Cruz Biotechnology, Dallas, TX, USA), or for 24 h with rabbit polyclonal Vesicular glutamate transporter 1 (vGLUT1, 1:500, Novus Biotechne, Littleton, CO, USA) and mouse monoclonal Vesicular gamma-aminobutyric acid (GABA) Amino Acid Transporter (vGAT, 1:250, Invitrogen, Carlsbad, CA, USA). The sections were then incubated for one hour with the secondary antibodies Alexa Fluor goat anti-rabbit 568 (1:500, Invitrogen) and Alexa Fluor goat anti-mouse 488 (1:500, Invitrogen). The slices were then washed with nuclear dye DAPI (1:1000, Sigma, Micanopy, FL, USA), mounted onto slides and cover slipped with Fluoromount (Thermo Scientific, Waltham, MA, USA). Images were taken with a BZ-X700 Fluorescent Microscope (Keyence, Osaka, Japan) at 40× magnification. We determined the colocalized presynaptic (SV2A) and postsynaptic (PSD95) puncta to indicate the proportion of active, connected synapses in the CA1 region of the hippocampus. Quantification of % colocalized was completed by counting (overlapping SV2A and PSD95 puncta)/(total SV2A and PSD95 puncta).

Given the sample size of *n* = 7 and *n* = 9 in each group, an expected effect size of 1.5 in the object recognition test, determined based on previous work [21], and *p* < 0.05, we were able to achieve a power of 0.79 for this preliminary analysis (G*Power 3.1, [22]). Data were first tested to confirm if they met assumptions of normality as assessed by examining the skewness, kurtosis and the Shapiro–Wilk value of the calculated residuals. Student’s *t*-tests were used to compare SHAM to TMS, and between five- and ten-minute-time-point paired *t*-tests were used. We tested for violations of homogeneity of variance using Levene’s statistic. If the homogeneity of variance was violated, significance was determined with the Welch Test, this case is noted in the results. All statistics were performed using SPSS (version 22, IBM, Armonk, NY, USA) and graphs produced using GraphPad (version 7, Prism, San Diego, CA, USA).

## 3. Results

As rTMS was targeting the cerebellum, it is important to assess whether the stimulation affected the ambulatory abilities of the animals which the cerebellum may have been involved in. We found no difference in the locomotor abilities (students *t*-test, *p* > 0.05) between the TMS and SHAM mice as assessed with OFT. For the ORT, there was no difference between the SHAM and TMS mice in the exploration time of the objects during the familiarization session (*p* > 0.05). At 5 min, the mice treated with rTMS had a greater recognition index than the SHAM mice (0.22 ± 0.09 vs. −0.06 ± 0.08, students *t*-test, *p* < 0.05). However, at 10 min, there was no difference due to the SHAM mice having improved their recognition by 0.13 index points from the 5 min time point (−0.06 ± 0.08 vs. 0.07 ± 0.06), paired *t*-test *p* < 0.05; Figure 1A). Interestingly, although there was no difference between the SHAM and rTMS mice in the total exploration time of the objects at 5 min, at 10 min the SHAM mice had greater exploration than the TMS mice (86.16 ± 13.81 vs. 51.70 ± 6.29, students *t*-test, *p* < 0.05; Figure 1B). Finally, there was no difference in the mice treated with TMS or SHAM in the time spent in the open areas on the O-maze (Student’s *t*-test, *p* > 0.05).

We explored the colocalization of SV2A, a marker of synaptic vesicles in the pre-synapse of both glutamatergic and GABAergic neurons with PSD95, a scaffolding protein in post-synapse of glutamatergic neurons. Colocalization of these puncta indicate active glutamatergic synapses in the CA1 [21]. rTMS stimulation to the cerebellum increased the colocalization of SV2A and PSD95 in the CA1 to 25.44% (±0.04), compared to 12.42% (±0.02) for the SHAM mice (Welch’s *t*-test, *p* < 0.05; Figure 2A). To examine the overall balance of glutamatergic and GABAergic neurons, we next determined the presence of neurons containing vGLUT or vGAT in the frontal cortex. vGLUT is a marker of vesicular glutamate transporters expressed in glutamatergic neurons, and vGAT is a marker of vesicular GABA transports expressed in GABAergic neurons. There was no difference between mice treated with rTMS and SHAM (Figure 2C,D).

## 4. Discussion

In this study, we have shown preliminary evidence that hindbrain-targeted rTMS improves the efficiency of discrimination of the object recognition test. The mice treated with hindbrain rTMS were able to achieve discrimination between the novel and familiar objects in the first five minutes of the experimental protocol, with less exploration of the objects, whilst the SHAM mice took longer to improve their discrimination. This improvement in discrimination efficiency was associated with an increased number of functional synaptic connections in the hippocampus. In humans, cerebellum dysfunction has been associated with more time to complete a learning task, irrespective of motor difficulties [23], and fronto-cerebellar connectivity has been shown to be involved in processing speed for cognitive tasks [24]. Additionally, in primate studies, cerebellar lesions significantly reduced response times in a working memory task [25]. Therefore, there is strong evidence that targeting stimulation to the hindbrain and cerebellum could affect components of cognition that would improve the efficiency of memory functions. Future studies should employ measures that directly investigate these specific components, such as attention and processing speed, or a complex episodic memory task to examine the efficiency of the integration of multiple memory modalities [26].

Previous modelling work has shown that lambda coil placement would target visual cortical regions, the superior colliculus and lobules IV/V of the cerebellum [19]. While we do not expect that targeting the visual cortex or superior colliculus would have any substantial effects on emotional and cognitive processing, previous work on the anterior cerebellum has confirmed its involvement in cognition in mice, but not in emotional processing [27]. Studies have demonstrated that manipulations to specific areas even within this cerebellum lobule can have varying impacts on recognition tests. In one study, a lesion of the lateral area of the lobule reduced exploration in the object recognition test; however, chemogenetic excitation of Purkinje cells within in the same area had no effect [27]. In contrast, in a different study, optogenetic stimulation of the medial area of the lobule did not affect performance in the object recognition test but did disrupt the spatial recognition test [20].

This current study also showed biochemical evidence that rTMS stimulation of the hindbrain altered hippocampal function. Unfortunately, we cannot distinguish with the experimental design if the increased synaptic connectivity of the hippocampus was due to direct alteration from the stimulation or due to the stimulation increasing the ability of the mice to perform the memory task, which then would induce greater hippocampal connections [21]. There is strong evidence to suggest a functional link between the hippocampus and cerebellum [3]. Although there has been little evidence that a direct pathway exists, secondary pathways through the thalamic nucleus, reticular nucleus and locus coeruleus have been found. Additionally, specific manipulations of the lobule IV/V are known to alter hippocampal signaling [20]; therefore, rTMS targeted to the cerebellum could be an alternative pathway for targeting this important brain structure.

GABAergic signaling is important in mood disorders, unfortunately we did not see a change in the balance of GABA and glutamate in the frontal cortex; this is in line with the lack of anxiety changes within these mice. These were healthy, young mice and rTMS often only shows an effect in a situation of deficit. Future studies could employ a model to induce an anxiety-like phenotype within the mice to explore if cerebellar stimulation would have an anxiolytic effect. Secondly, as previously mentioned, the lobules IV/V have been found to have little functional effect on other mood disorder measures in mice, so potentially other cerebellar targets could be explored for future studies.

In conclusion, the current study provides additional evidence supporting the role of the cerebellum in cognition. More importantly, it provides preliminary evidence that through the use of rTMS, the upstream effects of neurostimulation can induce stronger structural and functional connections to areas such as the hippocampus. Although there was no alteration of GABA and glutamate neuronal signaling, more optimal experimental conditions should be explored as this is counter to some of the existing literature. As the effects of rTMS are more prominent in the recovery of dysfunction, future research should build upon these preliminary findings to examine the effect of this hindbrain treatment protocol in models of disorders, especially in those where there is evidence of cognitive and cerebellar dysfunction.

## Figures and Tables

**Figure 1 brainsci-13-01425-f001:**
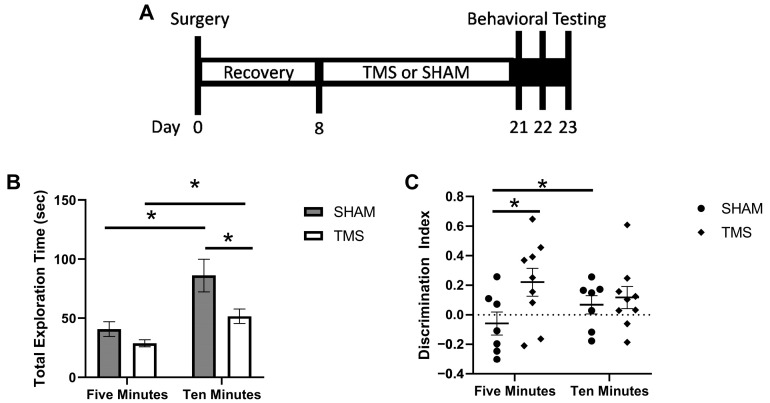
(**A**) timeline of experimental events. (**B**) discrimination index for the first 5 min and full 10 min of the object recognition test. (**C**) total exploration time of both familiar and novel objects in the testing phase of the object recognition test. * *p* < 0.05.

**Figure 2 brainsci-13-01425-f002:**
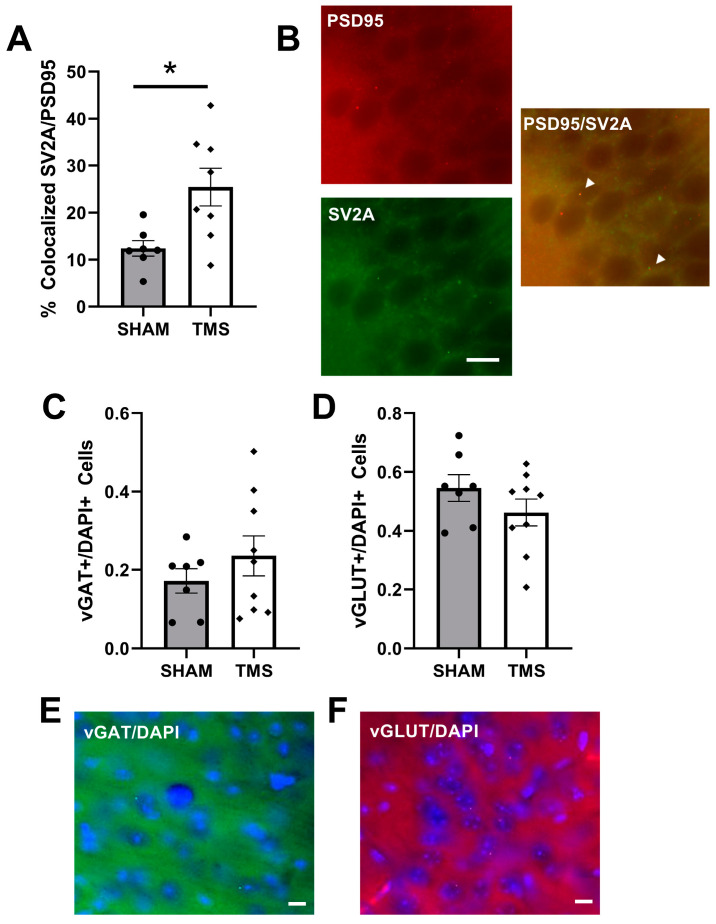
(**A**) percentage of total puncta which had colocalized Synaptic vesicle glycoprotein 2A (SV2A) and Postsynaptic density protein 95 (PSD95) in the Dorsal CA1 region of the hippocampus. (**B**) representation of colocalized SV2A/PSD95 puncta in the CA1 shown with white arrows. (**C**) vesicular GABA Amino Acid Transporter (vGAT) normalized to total number of cells in the frontal cortex. (**D**) vesicular glutamate transporter 1 (vGLUT) normalized to the total number of cells in the frontal cortex. (**E**) example of vGAT staining in frontal cortex. (**F**) example of vGLUT staining in frontal cortex. * *p* < 0.05, bars represent 10 µm.

## Data Availability

Data are available from author upon request.

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
