# Peer review of "Hindbrain Stimulation Modulates Object Recognition Discrimination Efficiency and Hippocampal Synaptic Connections"

_brainsci, 2023, doi:10.3390/brainsci13101425_

Round 1
Reviewer 1 Report
Specify the method Determining the sample size
Were there any worsening or no effect in the treatment group?
Describe in detail SV2A, PSD95, vGLUT or vGAT and GABA ergic neurons
The results should detail the percentage change in scores between the treatment group and the sham group.
Could you provide some examples of the slides you used to determine colocalized presynaptic (SV2A) and postsynaptic (PSD95).
Can you add examples of microscopic slides for neurons containing vGLUT or vGAT in the frontal cortex.
How would you explain that SHAM mice improved their recognition index at 10 minutes compared to a 5-minute time point ?
In conclusion it is impossible to relate the obtained results to the treatment of s Alzheimer’s disease and depression.
Reference is not formatted according to journal rules
Author Response
We would like to thank the reviewer for their time and comments on the paper. We believe your suggestions have greatly improved the manuscript.
As requested we have now included in the results section the actual mean values +/- standard error of the mean to indicate the differences between these two groups, as well as the difference between the 5 and 10 minutes time points, in the results. We have also included example images of both the PSD95/SV2A stain in the CA1 where we have shown an example of a colocalization of the counted puncta and example images of the vGLUT/DAPI and vGAT/DAPI stains in the frontal cortex and descriptions about the different markers we used and the neuronal subtypes these represent. And we have checked and reformatted all the references in the paper to the MDPI guidelines.
We would like to address the remaining points individually:
1. Specify the method Determining the sample size
Based on our previous research we had found three days of frontal TMS stimulation improved the object recognition discrimination index compared to SHAM with a cohens d of 1.29 (Heath et al 2021). Given the increase in amount of TMS (3 vs 14 days) we expected a greater effect size of TMS and therefore for the preliminary analysis we used a cohens d of 1.5, and power of 0.8 to determine that an appropriate group size would be n=9. Unfortunately, we were not able to fulfil this number in the SHAM group but due to the lack of intervention, this group had less variability and therefore we were still able to achieve enough power to show the effects of hindbrain TMS seen in this paper.
2. Were there any worsening or no effect in the treatment group?
In the group treated with TMS there was a decrease in their discrimination index from 5 minutes to 10 minutes but this change was not statistically significant (paired t-test, p>0.05).
3. How would you explain that SHAM mice improved their recognition index at 10 minutes compared to a 5-minute time point ?
We believe this result demonstrates a slower processing of episodic information that the SHAM mice previously received in the exploration phase, where it took these mice longer to discern the differences between the objects to determine the novelty and hence they had a greater interest in exploring the novel object more later in the testing phase. This could also be seen in their overall exploration: SHAM mice explored more in the second 5 minutes than the first 5 minutes, and comparing the exploration of Familiar and Novel objects from 5 to 10 minutes independently they explored Novel objects 68% more seconds compared to 5 minutes but 4% less seconds for the familiar objects.
4. In conclusion it is impossible to relate the obtained results to the treatment of s Alzheimer’s disease and depression.
We have reworded our conclusion so that it is more aligned with the results shown in this paper. We initially chose to make these conclusions as rTMS is an already prescribed treatment for depression and the increased interest for treating cognitive decline in Alzheimer’s disease with rTMS but we do understand that the results from this paper cannot be extrapolated to those uses.
Reviewer 2 Report
One crucial aspect to address in the introduction of this study is discussing how your work distinguishes itself from previously published studies. It is essential to highlight the unique contributions and novel elements that set your research apart from existing literature within the field. This comparison will provide a clear context for understanding why your study's approach, methodology, or findings offer valuable insights beyond what has already been explored by others. By establishing these points of differentiation early on, readers can appreciate the originality and significance of your work within the broader academic discourse surrounding this topic area.
Include a visual representation in the methods section that illustrates the methodology employed.
As mentioned throughout the study, this is considered a preliminary investigation. Therefore, it would be beneficial to outline the future directions for expanding upon this research.
Author Response
We thank the reviewer for their comments. We have adjusted our manuscript based on your recommendations and have outlined how we have addressed your specific points below.
1. One crucial aspect to address in the introduction of this study is discussing how your work distinguishes itself from previously published studies. It is essential to highlight the unique contributions and novel elements that set your research apart from existing literature within the field. This comparison will provide a clear context for understanding why your study's approach, methodology, or findings offer valuable insights beyond what has already been explored by others. By establishing these points of differentiation early on, readers can appreciate the originality and significance of your work within the broader academic discourse surrounding this topic area.
We have added an additional paragraph in our introduction to highlight the novelty of our work in the field. There is increased evidence of the cerebellum-hippocampal connection's importance for cognition but whether we can use brain stimulation to activate this pathway was still unknown. We believe this research has added significance to this field to not only provide additional evidence that this cerebellar-hippocampal pathway exists but that we can manipulate it with rTMS.
2. Include a visual representation in the methods section that illustrates the methodology employed.
We have now included a timeline in figure 1 that shows the different experimental events.
3. As mentioned throughout the study, this is considered a preliminary investigation. Therefore, it would be beneficial to outline the future directions for expanding upon this research.
In our conclusion we now highlight that this research could be expanded by investigating in models of cerebellar dysfunction. Within our lab we hope to investigate the effect of cerebellar rTMS in a model of ageing which has shown cerebellar deficits which have been linked to cognition.
Round 2
Reviewer 1 Report
The work is of great practical and scientific interest. The topic is very relevant, especially considering the development of experimental medicine in studying of cognitive disorders and in the use of electrotherapy as treatment of these disease in future.
The purpose of the study is described clearly.
The rational for selecting that particular statistic, and which variables were entered into the statistic are described.
Statistical results are presented in a Table or Figure.
The title of the article matches the content. The purpose and objectives of the work are fully realized.
Discussion and conclusions follow logically from the results of the study and are fully consistent with the purpose of the study.
The main findings as related to the overall purpose of the study are discussed and explained in detail.
Conclusions is directly related to the data that was collected and analyzed.
However, there remains one important unresolved issue. Information on sample size determination should be included in the Materials and Methods. In addition, the small number of participants in sham group may affect the validity of the results. Perhaps if you reduce the power of your study to 70% or 60%, the minimum number of patients in the groups will decrease
Author Response
We thank the reviewer for their time reviewing the manuscript again.
We apologize for not including of the power analysis in the manuscript. This came from a misinterpretation of the previous comment that it was just requesting clarity on the calculation and not a request to include it in the manuscript. We have adjusted the manuscript now to include the power calculation and have adjusted the power to 70%, as suggested, which recalculated to n=7 per group.